# Redesigning Mathematical Curriculum for Blended Learning

**Gerry Stahl**

College of Computing and Informatics, Drexel University, Philadelphia, PA 19104, USA; Gerry@GerryStahl.net

**Abstract:** The Coronavirus pandemic has thrown public schooling into crisis, trying to juggle shifting instructional modes: classrooms, online, home-schooling, student pods, hybrid and blends of these. This poses an urgent need to redesign curriculum using available technology to implement approaches that incorporate the findings of the learning sciences, including the emphasis on collaborative learning, computer mediation, student discourse and embodied feedback. This paper proposes a model of such learning, illustrated using existing dynamic-geometry technology to translate Euclidean geometry study into collaborative learning by student pods. The technology allows teachers and students to interact with the same material in multiple modes, so that blended approaches can be flexibly adapted to students with diverse preferred learning approaches or needs and structured into parallel or successive phases of blended learning. The technology can be used by online students, co-located small groups and school classrooms, with teachers and students having shared access to materials and to student work across interaction modes.

**Keywords:** dynamic geometry; group practices; CSCL; group cognition; learning pods

## 1. Introduction: Student Pods during the Pandemic

Alternatives to the traditional teacher-centric physical classroom suddenly became necessary during the coronavirus pandemic to cover a variety of shifting learning options at all age levels. Although the creation of student "pods" (small groups of students who study together) was popularized as a way of restricting the spread of virus, it was rarely transferred to the organization of online learning as collaborative learning.

Research in the learning sciences has long explored pedagogies and technologies for student-centered and collaborative learning [1]. However, the prevailing practice of schooling has changed little [2]; students, parents, teachers, school districts and countries were poorly prepared for the challenges of the pandemic. Case studies from countries around the world documented the common perceptions by students, teachers and administrators of inadequate infrastructure and pedagogical preparation for online learning [3,4].

An abrupt rush to online modes found that the digital divide that leaders had promised to address for decades still left disadvantaged populations out [5,6]. Income inequality by class and nation correlates strongly with lack of computer and Internet access. In addition to confronting these hardware issues and low levels of computer training, teachers everywhere had access to few applications designed to support student learning in specific disciplines. They had to rely on commercial business software like Zoom and management systems like Blackboard, which incorporated none of the lessons of learning-sciences research.

While school districts planned for "reopening," administrators prepared scenarios for combining in-class, online, home schooling and small student pods. The plans kept shifting and little was done to prepare and support teachers to teach in these various combinations of modalities. Moreover, teachers were rarely guided in redesigning their curriculum for online situations, in which they were often neither trained nor experienced.

Pundits and early surveys were quick to call the attempt to teach online a failure and declare that it simply highlighted how important social interaction was to students. They

argued that online media severely reduced student motivation by removing inter-personal interaction [7,8].

However, the field of computer-supported collaborative learning (CSCL) has always emphasized the centrality of social interaction to learning, demonstrating that sociality could be supported online as well as face-to-face [9,10]. Micro-analyses of knowledge building in CSCL contexts detail the centrality of social interaction to effective online collaborative learning and even the students' enjoyment of the online social contact [11]. The source of asocial feelings is the restriction of online education to simply reproducing teacher lectures and repetitive individual drill. It is necessary to explicitly support social contact and interaction among students to replace the subtle student-to-student contact of co-presence. This can be done through collaborative learning, which simultaneously maintains a focus of the interaction on the subject matter.

The pandemic forced teachers to suddenly change their teaching methods and class-room practices, as reported by [12]. The sudden onset of pandemic conditions and school lockdown made it infeasible to introduce new technologies, let alone scale up research prototypes for widespread usage. Nevertheless, the lessons of the pandemic should lead over the longer run to more effective online options, as well as preparation in terms of infrastructure, support, attitude and skills for innovative online educational approaches and applications [13].

In the face of the pandemic, teachers and school districts were largely on their own to adapt commercially established technologies like Zoom and Blackboard to changing local circumstances. One innovative example was an attempt to make teacher presentations in Blackboard more interactive by instituting a hybrid audience of some students in class (to provide feedback to the teacher) and others online [14]. Other researchers stressed the need to go further and introduce an intermediate scale between the individual students and the teacher-led classroom—namely a student-centered small-group or pod learning unit [15]. The following provides an example of how a careful integration of existing technologies (Zoom or Blackboard with GeoGebra) can support pod learning and blend the online with in-class as well as the small group with whole classroom.

This article describes how the Virtual Math Teams (VMT) research project translated the ancient pedagogy of Euclidean geometry into a model of CSCL, and how that was then further redesigned to support blended-learning pedagogy for pandemic conditions (with GeoGebra Classes). This can serve as a prototype for the blended teaching of other subjects in mathematics and other fields. If such a model can succeed during the pandemic, it can herald on-going practical new forms of education for the future. The pandemic experience will change schooling to take increased advantage of online communication and offers an opportunity for CSCL to guide that process in a progressive direction. The approach described here using GeoGebra Classes with VMT curriculum can be implemented immediately, during the pandemic, and then further developed later for post-pandemic blended collaborative learning.

## 2. Designing for Virtual Math Teams

The VMT research project was conducted at the Math Forum at Drexel University in Philadelphia, USA, from 2004 through 2014. The VMT research has been documented in five volumes analyzing excerpts of actual student interaction from a variety of viewpoints and methodologies [11,16–19].

The project was an extended effort to implement and explore a specific vision of computer-supported collaborative learning (CSCL), applied to the learning of mathematics:

- First, it generated and collected data on small online groups of public-school students collaborating on problem solving.
- Second, it provided computer support, including a shared whiteboard and a dynamic-geometry app.
- Third, it analyzed the group interaction that unfolded in the team discourse.

- Fourth, it elaborated aspects of a theory of "group cognition" [19]. Several papers published during this period and contributing to the broad vision of CSCL have now been reprinted and reflected upon in *Theoretical Investigations: Philosophic Foundations of Group Cognition* [11]. Several chapters in this volume analyze aspects of group cognition based on excerpts of student discourses during VMT sessions.

The VMT project cycled through many iterations of design-based research (design, trial, analysis, redesign), developing an online collaboration environment for small groups of students to learn mathematics together. The eleven chapters of [17] describe the project from different perspectives: the CSCL vision; the history, philosophy, nature and mathematics of geometry; the theory of collaboration; the approach to pedagogy, technology and analysis; the curriculum developed; and the design-based character of the research project. The theory of group cognition provides a framework for pod-based education by describing how knowledge building can take place through small-group interaction, with implications for conceptualizing collaborative learning, designing for it, analyzing group-learning processes/practices and assessing its success. The theory explores the inter-weaving of individual, group and classroom learning.

The VMT software eventually incorporated GeoGebra (https://www.geogebra.org (accessed on 31 March 2021)), an app for dynamic geometry, which is freely available and globally popular (available in over 65 languages). Dynamic geometry is a computer-based version of Euclidean geometry that allows one to construct figures with relationships among the parts and then allows the constructed points to be dragged around to test the dependencies-providing immediate visual feedback [20–22].

As part of the VMT Project, curricular units were designed and tried out in online after-school settings (primarily in the Eastern USA), with teacher training on how to guide the student groups and how to integrate and support the online collaborative learning with teacher presentations, readings, homework and class discussion [23]. The geometry activities provided hands-on experience exploring the basics of dynamic geometry in small-group collaboration. Student peer discussion was encouraged that would promote mathematical discourse and reflection [24]. In this way, the research project translated Euclid's curriculum into the computer age. Euclid's *Elements* [25], which had inspired thinkers for centuries, was reworked in terms of dynamic geometry and a learning-sciences perspective [2].

## 3. Redesigning for Pandemic Pods with GeoGebra Classes

The VMT platform was no longer available when the pandemic appeared and made the need for supporting online learning particularly urgent. While teachers and students can download GeoGebra without VMT, that would not support full collaboration, where several students can work together on a shared geometric figure. Fortunately, GeoGebra recently released a "Class" function, in which a teacher can invite several students (a pod) to work on their own versions of the same construction, and the teacher can view each student's construction work and discussion in a Class dashboard (Figures 1 and 2). The dashboard provides a form of "learning analytics" [10] support for the teacher, which can also be adapted to facilitate student collaboration.

To take advantage of GeoGebra Classes, VMT's dynamic-geometry curriculum has now been adapted to small pods or even home-schooled individual students using the Classes functionality. The new curriculum is called *Dynamic Geometry Game for Pods* [26]. Using a set of 50 GeoGebra activities that cover much of basic high-school or college geometry, the instructions and the reflection questions were reworked for the new scenario (Figures 3 and 4). The sequencing of tasks was maintained from VMT, which roughly followed Euclid's [25] classic presentation as well as contemporary U.S. Common Core guidelines for geometry courses [27].

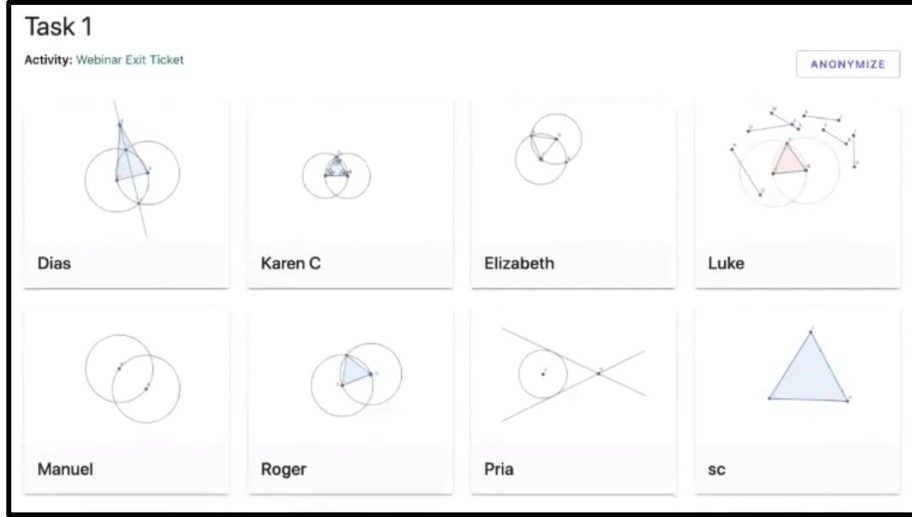

**Figure 1.** The GeoGebra Class dashboard displays the current state of each student's work on a selected task. In this example, the students are learning Euclid's construction of an equilateral triangle.

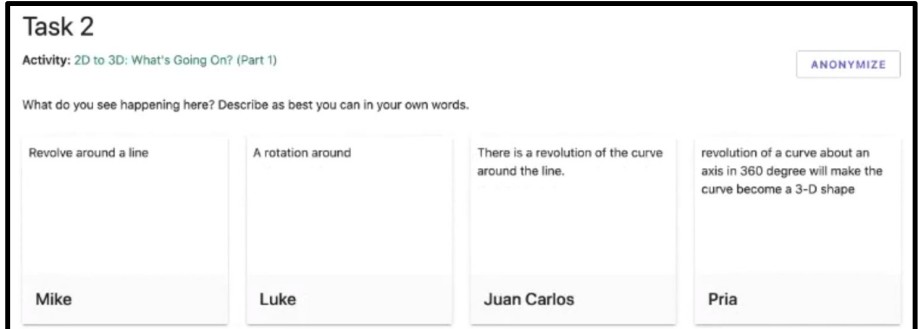

**Figure 2.** The GeoGebra Class dashboard also displays each student's response to selected questions. In this example the students are discussing rotating a 2-D curve into the 3rd dimension.

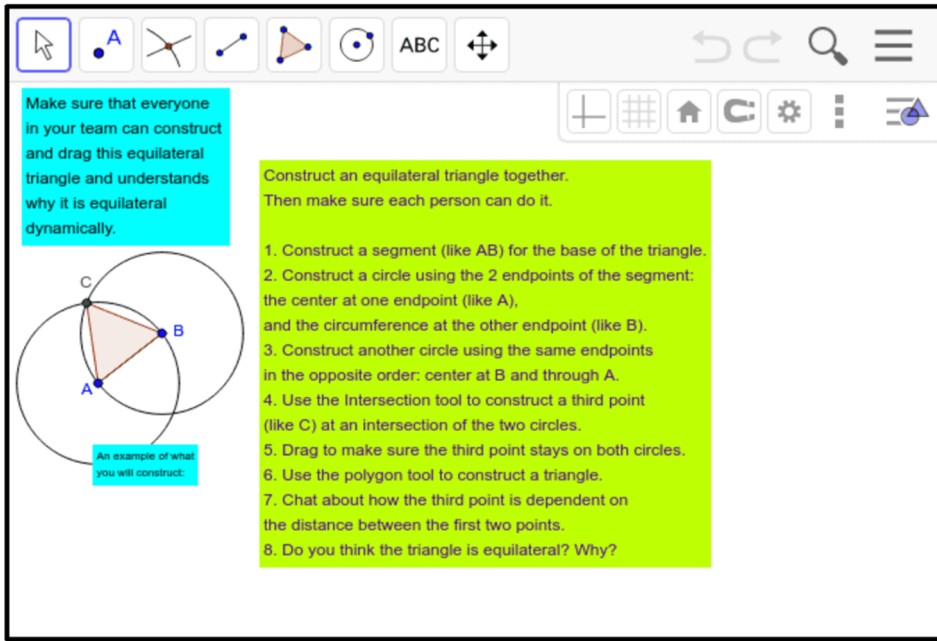

**Figure 3.** One of 50 tasks for student pods: Euclid's construction of an equilateral triangle.

> **Questions.**
>
> Did you construct your own equilateral triangle?
>
> Did you use the DRAG TEST to make sure it works properly?
>
> The equilateral construction opens up the world of geometry; if you understand how it works deeply, you will understand much about geometry.
>
> In geometry, a circle is defined as the set of points that are all the same distance from the center point. So every radius of a certain circle is the same length.
>
> Drag each point in your triangle and discuss how the position of the third point is dependent on the distance between the first two points.
>
> Is your triangle equilateral (all sides equal and all angles equal)?

**Figure 4.** A set of reflection questions for members of pods to discuss related to the task in Figure 3.

The revised curriculum is available on the GeoGebra repository site as an interactive GeoGebra book (https://www.geogebra.org/m/vhuepxvq#material/swj6vqbp (accessed on 31 March 2021)). Additionally, a free e-book is available so people can conveniently review the curriculum offline [26]. The book's introductions guide classroom teachers, home-schooling parents, pod tutors or self-guided students to use the curriculum. The format is that of a game with successively challenging levels, which must be conquered consecutively. It is structured as a sequence of five parts, each including about 10 of the hour-long curricular activities, grouped by geometry level and degree of expertise required. The game levels are: (1) beginner, (2) construction, (3) triangles, (4) circles, (5) dependency, (6) compass, (7) congruence, (8) inscribed polygons, (9) transformation, (10) quadrilaterals, (11) advanced geometer, (12) problem solver and (13) expert.

The ideal usage would be by pods of students working online and communicating through the dashboard. A pod coordinator or teacher can provide all participants with access to the real-time dashboard, so that everyone can observe and discuss what everyone else is doing in GeoGebra and typing in the Class interface. Furthermore, GeoGebra can be shared in Zoom, to provide spoken interaction and recording of sessions for student reflection, teacher supervision or researcher analysis.

Note that the Class functionality is not fully collaborative, even when all students have access to the dashboard. Individual students work in their own construction areas (Figure 1), unlike the shared workspace of the VMT software (Figure 3). Additionally, each student answers the reflection questions in their own window (Figure 2), rather than in an interactive chat window as in VMT. However, at least the students can see each other's work and learn from it. Moreover, if GeoGebra is embedded in Zoom, then the students can verbally discuss their approaches together. The limited support for collaboration is a trade-off of using established software for innovative pedagogy.

The goal is that math teachers and others can adapt the use of this curriculum and technology to diverse and rapidly changing teaching conditions and learning modalities. If used with full online access—including the Class dashboard shared by everyone, possibly embedded in Zoom—the collaborative learning experience can approach that envisioned in the VMT research. However, it can also be used in other ways and across various presentation modalities of blended approaches. Student work carried out individually can be shared within a Class pod and then presented in a whole classroom setting, whether virtual or face-to-face.

The usage of GeoGebra in a collaborative online session can provide all students with hands-on experience in geometry construction and investigation (manipulation and reflection). A major advantage of collaborative learning is that students can help each other, pooling their partially developed skills and understanding. However, it is also important for teachers to provide introductions to new ideas and to review in the classroom context the work that students are doing in pods or individually. Furthermore, individual

students must make sense of the material for themselves; reading and working individually on problems is important to support collaborative learning. That is why teachers should orchestrate blended learning, incorporating individual, small group and classroom learning in a coordinated, mutually supportive way. Of course, students learn best in diverse ways, so it is productive to offer them alternative educational modalities. Teachers can adapt and mix the modalities in response to local circumstances and learning differences among their students.

## 4. Findings from VMT Trials

The VMT Project was conceived and executed as extended design-based research (DBR), as detailed in [17]. This involved innovations in technology, pedagogy, assessment and theory. Each aspect of the VMT Project has been reviewed in multiple formats and contexts by international researchers from relevant disciplines.

Findings from the project have been discussed in about 250 publications, including peer-reviewed workshops, conference papers, journal articles, dissertations and books. The project evolved over a decade, prototyping and testing technologies and curricula that underwent multiple iterative revisions each year. The current curriculum for blended learning, *Dynamic Geometry Game for Pods,* is the latest iteration, moving from the VMT software platform to the GeoGebra Class function to support blended learning including collaborative learning in online student pods.

Although a variety of analysis approaches were applied to identify successes and problems during VMT trials, most of the published analyses used a form of conversation analysis adopted from informal conversation to the interaction of online school mathematics. While most of the analyses focused on brief interactions among small groups of students, some included longer sequences, sometimes spanning multiple sessions. For instance, the entire interaction of a group of three middle-school girls—the "Cereal Team"—was followed longitudinally across eight hour-long online sessions and was subjected to detailed micro-analysis of all the discourse and geometry construction [16,17].

As suggested by the title of [17], *Translating Euclid: Designing a Human-Centered Mathematics*, the pedagogy was converted away from expecting students to accept and memorize concepts, theorems and techniques based on authority. Instead, the project promoted a student-centered and inquiry-based approach of exploration, feedback and discourse based on situated and embodied interaction with computer-based artifacts and guided discussion practicing the use of mathematical terminology.

Although the VMT Project was originally intended to investigate and document phenomena of *group cognition* [19], in the end it proposed a methodological focus on *group practices* [16]. The sequencing of challenges in the *Dynamic Geometry Game for Pods* is carefully designed to guide student groups and individuals to adopt group practices and individual skills needed to progress through the process of collaboratively learning dynamic geometry. For instance, procedures for placing lines, dragging points, constructing circles and checking connections among objects are practiced before more complex constructions are proposed, which rely on these skills. The VMT research indicates that such an approach can be effective without being overly directive if a group of students can explore and discuss each technique collaboratively. The *Dynamic Geometry Game for Pods* is based on this body of findings, as well as on the extensive learning-science literature that underlies the VMT project's theory of group cognition, reviewed in [11].

## 5. Supporting Group Practices in Blended Learning

Teachers, parents and pod organizers can now use the GeoGebra book with its 50 challenges for courses in high-school geometry. Educators in other fields could follow this example and develop analogous curriculum and technology usage. Then, the results of such educational interventions could be collected, shared and analyzed. Analysis techniques honed during the VMT Project [28] could be used along with other methods to

investigate collaboration patterns in interaction discourse, the adoption of targeted group practices and advancement of learning goals.

This approach contrasts with the view of learning as primarily a psychological process of changing an individual's mental contents or cerebral representations [29,30]. Rather, individual learning is seen as largely a result of group and social processes or practices in which multiple people, artifacts, technologies and discourses interact to evolve cognitive products at the group level, such as geometric constructions, informal proofs, group reports and textual responses to questions [31]. Such group products require the establishment and maintenance of mutual understandings, intersubjectivity, distributed cognition, communal conceptualizations, common interpretations of problems, collaborative problem solving and shared knowledge. While individuals contribute to these group phenomena, the collective products have a life of their own [32–37].

One way that group cognition can result in individual learning is through the adoption of *group practices*, which then provide models for individual behavior [11] (chp. 16). For instance, a pod of students working on a geometry problem can encounter a concept, theorem or technique that may originate with a pod member, from the problem description or from the history of geometry. The pod discussion may then explicitly discuss what was encountered, come to a shared understanding of how it applies to the pod's current situation and even overtly agree to use it. In subsequent interactions, the pod simply applies the new practice without discussing it again. It becomes a tacit group practice, recognized by everyone in the pod. Pod members may also retain this practice as their own individual mathematical skill when they work outside the pod.

While the theory of group cognition and group practice has been discussed at length in the reports of the VMT Project, it will be interesting to see how these theories are manifested in new situations in which the *Dynamic Geometry Game for Pods* or analogous curricula are enacted. In addition to these quite broad theories, the VMT Project developed characteristics that may be more specific to digital geometry. It will be important to investigate the applicability of these features in new contexts and disciplines.

A central focus of the *Dynamic Geometry Game for Pods* is on the practices involving *dependency* as central to dynamic-geometric constructions. For instance, in constructing an equilateral triangle with radii of equal circles, it is essential that the length of the three sides are dependent upon the equal radii, even when a triangle vertex or a circle center is dragged to a new location. Indeed, the proof that the triangle is equilateral hinges on this dependency—and has for thousands of years since Euclid [25]. Viewing constructions in terms of practices that establish and preserve dependencies (rather than in terms of visual appearance or numeric measurements) is quite difficult for students to learn. One can observe such an insight as it emerges in the discourse of a pod, assuming that the curriculum has been effectively designed to promote such a group practice.

One aspect of curriculum design to support the adoption of specific group practices in dynamic geometry is to sequence tasks and associated practices carefully. This is clear in Euclid's carefully ordered presentation and in the hierarchies of theorems in every area of mathematics.

However, in collaborative learning of geometry, groups must adopt more practices than just the purely mathematical ones. Specifically, the micro-analysis of the eight sessions of the Cereal Team identified about sixty group practices that the group explicitly, observably enacted. These practices successively contributed to various core aspects of the group's abilities: to collaborate online; to drag, construct, and transform dynamic-geometry figures; to use GeoGebra tools; to identify and construct geometric dependencies; and to engage in mathematical discourse about their accomplishments.

Table 1 lists practices explicitly discussed by the Cereal Group and identified in the analysis of their discourse [16]. Each of these practices is illustrated in the commentary on the detailed transcript of the student group's interaction. One can see the group negotiating, adopting and reusing each group practice in the context of their mathematical problem solving and online collaborative learning.

**Table 1.** Identified practices adopted by the Cereal Group.

**Group Collaboration Practices:**

1. Discursive turn taking (responding to each other and eliciting responses).
2. Coordinating activity (deciding who should take each step).
3. Constituting a collectivity (e.g., using "we" rather than "I" as agent).
4. Sequentiality (establishing meaning by temporal context).
5. Co-presence (being situated together in a shared world of concerns).
6. Joint attention (focus on the same, shared images, words and actions).
7. Opening and closing topics (changing discourse topics together).
8. Interpersonal temporality (recognizing the same sequence of topics, etc.).
9. Shared understanding (common ground).
10. Repair of understanding problems (explicitly fixing misunderstandings).
11. Indexicality (referencing the same things with their discourse).
12. Use of new terminology (adopting new shared words).
13. Group agency (deciding what to do as a group).
14. Sociality (maintaining friendly relations).
15. Intersubjectivity (sharing perspectives).

**Group Dragging Practices:**

1. Do not drag lines to visually coincide with existing points, but use the points to construct lines between or through them.
2. Observe visible feedback from the software to guide dragging and construction.
3. Drag points to test if geometric relationships are maintained.
4. Drag geometric objects to observe invariances.
5. Drag geometric objects to vary the figures and see if relationships are always maintained.
6. Some points cannot be dragged or only dragged to a limited extent; they are constrained.

**Group Construction Practices:**

1. Reproduce a figure by following instruction steps.
2. Draw a figure by dragging objects to appear right.
3. Draw a figure by dragging objects and then measure to check.
4. Draw a figure by dragging objects to align with a standard.
5. Construct equal lengths using radii of circles.
6. Use previous construction practices to solve new problems.
7. Construct an object using existing points to define the object by those points.
8. Discuss geometric relationships as results of the construction process.
9. Check a construction by dragging its points to test if relationships remain invariant.

**Group Tool-Usage Practices:**

1. Use two points to define a line or segment.
2. Use special GeoGebra tools to construct perpendicular lines.
3. Use custom tools to reproduce constructed figures.
4. Use the drag test to check constructions for invariants resulting from custom tools.

**Group Dependency-Related Practices:**

1. Drag the vertices of a figure to explore its invariants and their dependencies.
2. Construct an equilateral triangle with two sides having lengths dependent on the length of the base, by using circles to define the dependency.
3. Circles that define dependencies can be hidden from view, but not deleted, and still maintain the dependencies.
4. Construct a point confined to a segment by creating a point on the segment.
5. Construct dependencies by identifying relationships among objects, such as segments that must be the same length.
6. Construct an inscribed triangle using the compass tool to make distances to the three vertices dependent on each other.
7. Use the drag test to check constructions for invariants.
8. Discuss relationships among a figure's objects to identify the need for construction of dependencies.
9. Points in GeoGebra are colored differently if they are free, restricted or dependent.
10. Indications of dependency imply the existence of constructions (such as regular circles or compass circles) that maintain the dependencies, even if the construction objects are hidden.

Table 1. *Cont.*

| Group Dependency-Related Practices: |
| --- |
| 11.  Construct a square with two perpendiculars to the base with lengths dependent on the length of the base. |
| 12.  Construct an inscribed square using the compass tool to make distances on the four sides dependent on each other. |
| 13.  Use the drag test routinely to check constructions for invariants. |

| Group Practices Using Chat and GeoGebra Actions: |
| --- |
| 1.   Identify a specific figure for analysis. |
| 2.   Reference a geometric object by the letters labeling its vertices or defining points. |
| 3.   Vary a figure to expand the generality of observations to a range of variations |
| 4.   Drag vertices to explore what relationships are invariant when objects are moved, rotated, extended. |
| 5.   Drag vertices to explore what objects are dependent upon the positions of other objects. |
| 6.   Notice interesting behaviors of mathematical objects |
| 7.   Use precise mathematical terminology to describe objects and their behaviors. |
| 8.   Discuss observations, conjectures and proposals to clarify and examine them |
| 9.   Discuss the design of dependencies needed to construct figures with specific invariants. |
| 10.  Use discourse to focus joint attention and to point to visual details. |

| |
| --- |
| 11.  Bridge to past related experiences and situate them in the present context. |
| 12.  Wonder, conjecture, propose. Use these to guide exploration. |
| 13.  Display geometric relationships by dragging to reveal and communicate complex behaviors. |
| 14.  Design a sequence of construction steps that would result in desired dependencies. |
| 15.  Drag to test conjectures. |
| 16.  Construct a designed figure to test the design of dependencies. |

The design of curriculum for collaborative or blended learning can be motivated by the goal of promoting the adoption of specific group practices. The curriculum can, for instance, scaffold collaboration practices like turn taking to get all students in a group involved. Then, it can support discourse practices to help groups make their meanings explicit and shared.

Some of the listed group practices are specific to the collaborative learning of dynamic geometry with GeoGebra. Many are generally supportive of productive collaborative interaction and discourse. Each subject area will have its own central practices to be supported and mastered, as well as the more universal ones. It is instructive to see the special demands of dynamic geometry. In addition to the focus on construction of dependencies and the associated discourse of how different elements of a figure are dependent upon each other, the use of GeoGebra introduces further specific challenges. For instance, it was necessary to design the VMT technology to allow all group members to observe each other's construction sequences in detail as they unfolded in real time in the app, because the animation of those processes could be quite informative [38]. In addition, the immediate feedback afforded by GeoGebra—for instance when someone dragged a point and the whole construction changed, revealing what was and what was not dependent on that point—was crucial for group behavior, discourse and learning.

## 6. Broadening the Model for Blended Learning

The proposed use of GeoGebra Classes illustrates the adaption of existing technology to an educational innovation explored in research using a prototype that is not available for widespread use during the pandemic. While the GeoGebra Classes functionality does not fully support small groups to share a workspace for exploring geometric construction, it does provide an available platform for student pods working within a teacher-led classroom. Students in a pod can see each other's work in real time and can reflect upon it by answering questions that are integrated into the curriculum. The teacher can also follow all the student work and discourse and display this within a classroom context. Thus, blended learning is supported with online GeoGebra, individual construction and reflection,

small-group interaction and classroom presentation and discussion. The latest version of the online VMT curriculum is fully incorporated in a motivational game-challenge format. Optionally, the GeoGebra Class can be embedded in Zoom or Blackboard to support additional online and blended functionality.

The research that lies behind the VMT curriculum resulted in enumeration of group practices that are important to support for collaborative learning in its subject domain of dynamic geometry. Research reports developed the theory of group cognition, which describes how small groups can build knowledge collaboratively, in orchestration with individual learning and classroom instruction. They analyzed in considerable detail the nature of online mathematical discourse and problem solving, including how to support and analyze it.

These features of the VMT experience will need to be reconsidered in the design and analysis of support for blended learning in other subject areas, particularly to the extent that curriculum and technology diverge from dynamic geometry and GeoGebra. Just as the VMT project focused its curriculum on geometric dependencies as central to mastering dynamic geometry, efforts in other disciplines may target concepts that underlie their subjects, much as Roschelle's [39] early CSCL physics support app targeted the understanding of acceleration as core to learning Newtonian mechanics or an algebra curriculum might revolve around the preservation of equalities.

Dynamic geometry is just one area of mathematics covered by GeoGebra. The software supports all of school mathematics from kindergarten through junior college. It is available in most major world languages. Thus, a teacher, parent or student who masters dynamic geometry through the curriculum discussed here can go on to explore other areas of mathematics with this kind of computer support. Learning scientists can develop curriculum units for all ages in all countries following the model illustrated here by the *Dynamic Geometry Game for Pods*.

This is not to say that all instruction should be provided in a CSCL format. Collaboration can be particularly productive for exploring problems that are somewhat beyond the reach of individual students. Additionally, small-group collaborative learning is most effective in sessions that are orchestrated into sequences of individual, group and classroom activities that support each other [40,41]. Blended learning approaches can supplement collaborative learning with complementary instructional modes. For example, a teacher presentation and student readings can precede online peer interaction, which is followed up by classroom discussion and reporting. While teachers struggle to find effective approaches in flipped, hybrid and online classes, there is now a clear opportunity for moving CSCL ideas into widespread practice. Exploration of pod-based learning during the pandemic could lead to important innovations in post-pandemic blended, collaborative and online learning.

It is difficult to convert courses from in-class to online. Typically, much of the effort goes into designing the curriculum and student tasks in advance and instituting new procedures and expectations for the students. A culture of collaboration must be established in the classroom over time. For instance, grading should be redefined in terms of group participation and team accomplishments. It takes several iterations to work things out; in each course, it requires teacher patience while students adjust. Students must be guided to communicate with their collaborators and to let go of competitive instincts.

The model proposed here is not a panacea for the current crisis of schooling, but rather an indication of a potential direction forward, for the remainder of the pandemic and beyond. We need to overcome the digital divide, promote collaborative learning, develop educational technology for exploring many domains, train teachers in online teaching, redesign curriculum to make it flexible for shifting modes of schooling. If we do not do this, then the learning sciences will have missed an opportunity to promote new forms of collaborative, inquiry-based and computer-supported learning. Only by meeting this challenge can we avoid the looming destruction of public education and the resultant serious worsening of social inequity.

**Funding:** The new research reported in this paper received no external funding.

**Institutional Review Board Statement:** The new research reported in this paper did not involve human subjects.

**Informed Consent Statement:** Not applicable.

**Data Availability Statement:** Not applicable.

**Conflicts of Interest:** The author declares no conflict of interest.

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
