# Peer review of "Redesigning Mathematical Curriculum for Blended Learning"

_education, doi:10.3390/educsci11040165_

Round 1
Reviewer 1 Report
The paper is a timely one and an important one in regard of the pandemic and its impact on learning and teaching practices. The design of the paper and study is appropriate, although I would like to see a greater link in the literature, to two areas: teachers' practices in relation to technology adoption; and second, the instances where teachers' practices have changed, prior to the current circumstances. I would also like to see a wider discussion of the findings of this study, to the literature on instructional technology.
Author Response
Thank you for your positive appreciation of the paper.
You are absolutely correct that more reference to various literatures was needed. I conducted a review of the most recent literature analyzing the reaction of educators to the adoption of technology during the pandemic. I had originally relied on popular press reaction and now looked at the recently emerging surveys and analyses. I have added a summary of these and cited some papers from a variety of countries, including some that cite many more such papers. I also included reference to two very recent (2021) papers in Edu Sci that presented proposals for innovations in blended learning, and indicated how my proposal builds on theirs.
In addition, I added references to the forthcoming Handbooks on CSCL and on the Learning Sciences, which incorporate and point to much of the most important current research literature related to instructional technology related to this paper.
Most of this added literature review appears in the expanded first section of the paper.
Reviewer 2 Report
Redesigning Mathematical Curriculum for Blended Learning.
This is a "communication" which I take to account for some of the weaknesses that I point to below.
In short, the communication wishes to highlight the relevance of an existing body of design based research carried out under the auspices of "The Virtual Math Teams" project to the widespread challenge of redesign of education. In addition to being carried out as design based research, the work is considered a part of the CSCL tradition.
This paper sets out by indicating the challenge posed by the Covid-19 pandemic in relation to providing education. The challenges are undoubtedly mirrored in different parts of the world, but the North-American context of the setting is not mentioned explicitly. This such an explication could help make sense of "these hardware issues", mentioned p.1. They were unclear to this reader. In a similar vein, the paper makes reference the widespread idea of "a digital divide". I also think this deserves contextualization. Finally, on contextualization, the idea of "student pods" was initially unknown to me. I think the paper does well in emphasizing that there is more of an immediate challenge - there will be a period after school shut downs where lessons of the rush to online teaching will have to be drawn (rather than "pundits... quick to call the attempt to teach online a failure..." p. 1).
The communication recounts redesigning earlier research and designs (dynamic-geometry curriculum) to the use of Geogebra (including a "class" function). As such, it points to a potentially valuable resource for design for learning across a range of educational levels. Also, the authors imagine teachers using the authors' designs and methods to redesign other subject areas, by analogy (p.6). Based on analysis of student interaction, a range of practices are identified that aid students in collaborating online. This is potentially very useful. However, I feel some of the practices could be exemplified more, and / or be subject to be more definite description (e.g #3 "constituting a collectivity", (p.7) )
In addition to greater contextualization, I feel the paper would benefit from
- More clear presentation of the method they mention in conclusion. ("The model proposed" p. 10. Aspect of this model seems to come down to very general movement in educational theory and designs for learning (blended, CSCL). Other seem to rely on a version of game-based learning (p.4). It can be all of the those, but for guidance to educational designers, it particularly blended and game based learning deserves a bit more clarity re. "model proposed". This is crucial, as it concern how I am to begin thinking about redesigning.
- Clarifying the relevance of the theoretical construct "group cognition" for the task of redesigning.
- Omit reference to existing designs /teachers being pedantic (p.5, 6), unless this is more clearly established.
- Possibly omit reference to increasingly critized concept of learning style.
Finally, I feel Figure 2, student 3 deserves some comment, Are there no dashboard screendumps available with a bit more writing?
In conclusion, I think this communication points to a potentially valuable resource in both a pressing matter, but also in education in general; Also, I feel the authors can do a better job of helping the reader envisage the "analogy" or other use of it in redesigning courses or more widely, designs for learning.
- learning styles
Author Response
Your review was extraordinarily helpful to me in highlighting a number of important areas that needed expansion and clarification, as well as several other points that could be clarified. I probably changed or added about a third of the paper in response. I will try to acknowledge some of the points here.
You were perceptive and sympathetic to note the "communication" mode of the paper. It is not a report on new research findings, but a discussion of how findings from a particular research project (VMT) -- already extensively reported upon -- could point to an approach to blended learning responsive to the pandemic conditions.
It was invaluable to have your feedback about some terminology (perhaps even jargon) that was problematic. I have tried to explain and contextualize terms like "pod" and "digital divide" as well as eliminate terms like "pedantic" (I meant "didactic", but even that is prone to misunderstandings and unintended innuendo) and "learning style."
In response to Reviewer 1, I added some international literature from underdeveloped countries that indicate that the "digital divide" separates not only rich and poor families in developed countries but separates the different countries as well. However, this is another issue, beyond the current paper.
I incorporated your distinction between what could happen quickly during the pandemic and lessons for the post-pandemic situation.
The list of group practices are spelled out and illustrated in the cited 300-page longitudinal study from which they emerged in the student discourse. They are only intended to be illustrative in this paper. Nevertheless, since they are listed here, I expanded the particularly abstract names for the first group of practices to be more meaningful and descriptive.
Most importantly given the purpose of the paper, I tried to spell out what I saw as the model. It remains somewhat vague -- perhaps necessarily given that I hope people will come at it with very different needs (e.g., different academic subject domains and different technologies to work with). However, I hope I have now managed to make my proposal more clear here.
In terms of your more specific suggestions on wording, I believe I have adopted them all.
Finally, the figures were, of course, quite inadequate. They came from an early announcement of the Class functionality. I have replaced them with more meaningful figures of the dashboard. I have also discussed how they differ from collaboration support in VMT.
I feel that your concerns and suggestions led to a considerable improvement of the paper and I hope you would to some degree agree.
Reviewer 3 Report
The model is adapted to a new situation (Teaching Euclidean geometry). It is a new, interesting way of using the software in teaching other subject.
Author Response
Thank you for your interest in this model for teaching geometry -- as well as expanded to other subject domains.
I have significantly expanded the description of the "model" as a basis for application to other areas. I have also clarified how the version with GeoGebra Classes responds to the needs of education during the pandemic with a blended approach.